# Biomedical Applications of Lactoferrin on the Ocular Surface

**DOI:** 10.3390/pharmaceutics15030865

**Published:** 2023-03-07

**Authors:** Uxía Regueiro, Maite López-López, Rubén Varela-Fernández, Francisco Javier Otero-Espinar, Isabel Lema

**Affiliations:** 1Corneal Neurodegeneration Group (RENOIR), Clinical Neurosciences Research Laboratory (LINC), Health Research Institute of Santiago de Compostela (IDIS), 15706 Santiago de Compostela, Spain; 2Department of Surgery and Medical-Surgical Specialties, Faculty of Optics and Optometry, University of Santiago de Compostela (USC), 15705 Santiago de Compostela, Spain; 3Department of Pharmacology, Pharmacy and Pharmaceutical Technology, University of Santiago de Compostela (USC), 15705 Santiago de Compostela, Spain; 4Institute of Materials (iMATUS), University of Santiago de Compostela (USC), 15705 Santiago de Compostela, Spain; 5Galician Institute of Ophthalmology (INGO), Conxo Provincial Hospital, 15706 Santiago de Compostela, Spain

**Keywords:** drug delivery systems, dry eye, keratoconus, lactoferrin, ocular surface infections, topical ocular administration

## Abstract

Lactoferrin (LF) is a first-line defense protein with a pleiotropic functional pattern that includes anti-inflammatory, immunomodulatory, antiviral, antibacterial, and antitumoral properties. Remarkably, this iron-binding glycoprotein promotes iron retention, restricting free radical production and avoiding oxidative damage and inflammation. On the ocular surface, LF is released from corneal epithelial cells and lacrimal glands, representing a significant percentage of the total tear fluid proteins. Due to its multifunctionality, the availability of LF may be limited in several ocular disorders. Consequently, to reinforce the action of this highly beneficial glycoprotein on the ocular surface, LF has been proposed for the treatment of different conditions such as dry eye, keratoconus, conjunctivitis, and viral or bacterial ocular infections, among others. In this review, we outline the structure and the biological functions of LF, its relevant role at the ocular surface, its implication in LF-related ocular surface disorders, and its potential for biomedical applications.

## 1. Introduction

Lactoferrin (LF), also known as lactotransferrin, is a mammalian iron-binding glycoprotein belonging to the transferrin (TF) family. For the first time, LF was discovered in bovine milk [1], and its similarity to the structure of TF focused further studies on understanding its properties in iron absorption and microbial growth regulation by iron derivation. In addition to iron, LF has a high affinity for other metals, such as zinc, copper, or manganese ions, being an important regulator of metal homeostasis in exocrine fluids [2,3,4]. Related to their iron-binding properties, TF and LF have been recognized as a powerful team in controlling free iron levels in body fluids, providing a high antioxidant capacity that contributes to mitigating damaging events such as oxidative stress or inflammation. Additionally, the wide array of LF biological functions as an anti-inflammatory, immunomodulatory, antiviral, antimicrobial, and antitumoral protein brings out its potential as a preventive and therapeutic target in different diseases. Due to its importance in tear fluid, LF has been proposed for the treatment of several ocular surface conditions. In this line, the reinforcement of LF by topical ocular administration would be highly beneficial to enhance its action at the ocular surface, showing great potential for biomedical applications. This review provides molecular, functional, and integrative perspectives on LF, including a description of its structure, its biological properties, its relevant role on the ocular surface, its involvement in LF-related ocular surface disorders, and its potential for biomedical applications.

## 2. Lactoferrin Structure

Human LF has a molecular weight of approximately 80 kDa, with an amino acid sequence of 691 amino acids, quite similar to different mammalian species [5,6,7]. Its structure includes two homologous globular lobes (N-lobe and C-lobe) with two domains per lobe (N1, N2, C1, and C2) [8]. Both lobes have specific ligands with ideal chemical and geometrical properties for high-affinity binding to ferric ions (Fe^3+^) [9]. These ligands contain two tyrosine residues, one histidine, and one aspartic acid. During the iron-binding process, interaction of the four amino acid residues is required, so Asp60, Tyr92, Tyr192, and His253 covalently bind to the Fe^3+^ atom in the presence of one carbonate ion (CO_3_^2−^). Moreover, the two oxygen atoms of the carbonate ion also bind to Fe^3+^. Iron binding by the LF has been described as a cooperative binding process in which the N lobe binds the iron ion first and then promotes the C lobe to bind another Fe^3+^ ion.

Two conformational states of this protein can be characterized: holo-LF, as an iron-free protein (the two lobes, C and N lobes, are free of iron), and apo-LF in a fully iron-loaded protein mode (when the two lobes have one iron atom bonded), the last one stable and resistant to protease degradation [10,11]. The helical bonding of the LF lobes allows maintaining the binding with ferric ions, even at pH as low as 3.5, making LF a powerful iron scavenger that prevents the precipitation of the ions into insoluble hydroxides [9]. Changes in the structure of the LF iron domains promote the release of the ions. The main events that give rise to the structural changes responsible for iron release are: (I) the reduction of the ferric iron to the ferrous form (LF has a low affinity for the ferrous ions), (II) specific LF receptors (such as the LF receptor, mainly expressed in the intestinal epithelial cells), and (III) a pH decrease (which can induce the protonation of the carbonate ion, tyrosine, and histidine ligands, weakening the binding to iron and leading to its release) [12]. Figure 1 shows the secondary and tertiary structures of human LF.

## 3. Lactoferrin Biological Functions

LF’s multifaceted nature has aroused great interest in its potential as a therapeutic and pharmacological target. In this sense, its iron-binding ability turns it into an essential regulator of toxic-free iron levels in exocrine fluids. In addition, LF has shown anti-inflammatory, immunomodulatory, antioxidant, antimicrobial, antiviral, and anticancerogenic properties, playing a fundamental role in the host defense.

### 3.1. Lactoferrin and Iron Homeostasis: A Crucial Protein in Antioxidant Protection 

Iron is an essential element required for a wide range of cellular functions and pathways. However, iron homeostasis must be strictly controlled, because its dysregulation can lead to potentially cell-damaging events. Increased free iron levels catalyze some reactions (such as Fenton or Haber/Weiss reactions) that produce free radicals highly toxic and harmful to cells through their ability to damage lipids, proteins, and other cellular components [13]. 

LF also participates in the regulation of iron homeostasis through several pathways due to its iron-binding ability. Firstly, LF binds to Fe^3+^ ions, reducing the quantity of free or unbound iron available in the extracellular space. Moreover, LF can interact with ceruloplasmin (Cp), a ferroxidase that converts ferrous into ferric ions. In this regard, the direct transfer of ferric ions from Cp to LF avoids free iron circulation and, consequently, the activation of damaging reactions [14,15]. Therefore, LF is a key factor in regulating free iron and protecting underlying cells and tissues against free radicals and oxidative stress in the extracellular space.

Cellular iron uptake, storage, and export are controlled by strict mechanisms responsible for the proper balance of intracellular iron levels (Figure 2). There is evidence that LF can protect against intracellular iron overload by its anti-inflammatory action over the proinflammatory cytokine interleukin-6 (IL-6) [16,17,18]. The protection against iron overload derives from the effects of IL-6 on the expression of molecules involved in regulating the intracellular iron levels, such as ferroportin (Fpn), the main protein responsible for the export of ferrous iron from an intracellular to extracellular space. IL-6 induces an under-expression of Fpn, reducing the iron release and, consequently, promoting iron overload inside the cell. The anti-inflammatory activity of LF on IL-6 involves the downregulation of this cytokine expression, which stimulates Fpn activity and shields the cell from the iron overload [19].

In addition, specific receptors for LF have been identified on the surfaces of different types of cells and tissues, suggesting that LF could also participate in iron absorption, especially at the intestinal level [20]. 

### 3.2. Lactoferrin in the Immune and Inflammatory Response

LF is secreted by innate immune cells, epithelial cells, and some glands, supporting the presence of significant concentrations of LF in biological fluids and exocrine secretions that cover mucosal sites as gateways for pathogens. The active participation of LF in the modulation of innate immune responses has been confirmed in several ways through its ability to: (I) regulate T- and B-cell maturation, (II) increase natural killer (NK) cell activity, (III) block or inhibit the complement pathway, (IV) induce macrophage function from cytokine production, and (V) inhibit intracellular pathogen proliferation [21,22,23,24].

Immunological functions of LF are associated with its cationic charge, allowing its interaction and binding to negatively charged cells of the immune system, triggering signaling pathways and modulating cellular processes such as differentiation, migration, and proliferation. Beyond its functions from the extracellular space, LF can be internalized into the cell and transported to the nucleus, where it binds to the DNA and triggers different signaling cascades [25]. Different LF receptors have been in vitro characterized, including intelectin-1 (ITLN1), low-density lipoprotein receptor-related protein 1 (LRP1), toll-like receptors 2 (TLR2) and 4 (TLR4), cluster of differentiation 14 (CD14), asialoglycoprotein receptor (ASGPR), nucleolin, and cytokine receptor 4 [20,26,27,28,29,30]. ITLN1 is the receptor with the highest affinity for LF [31]. It is expressed by intestinal and biliary epithelial cells and promotes LF uptake and internalization, probably contributing to iron absorption. Moreover, ITLN1 recognizes microbial carbohydrate chains in a calcium-dependent manner, playing a defensive role against microorganisms. LRP1 is a low-specific LF transmembrane receptor expressed by neurons, hepatocytes, smooth muscle cells, skin keratinocytes, and fibroblasts. LRP1 is involved in multiple processes such as the endocytosis and phagocytosis of apoptotic cells, lipid homeostasis, kinase-dependent intracellular signaling, and β-amyloid precursor protein (APP) metabolism, as well as neuronal calcium signaling and neurotransmission. LF binding to LRP1 promotes and modulates several LRP1 functions and induces the activation of processes such as mitogenesis in osteoblasts or keratinocyte proliferation and migration [31]. In this regard, LF’s target molecules, cells, and receptors directly control certain biological functions. Likewise, by the TLRs and NF-kB pathway, LF acquires the ability to modulate the inflammatory response, preventing and reducing the release of some proinflammatory proteins (including IL-6, IL-1B, and IL-8); promoting the expression of anti-inflammatory cytokines such as IL-4 or IL-10; and regulating the activity of the complement. Its anti-inflammatory power is highlighted as one of its most interesting pharmacological properties [24,32,33,34]. 

### 3.3. Antibacterial Activity

The Fe^3+^ uptake capacity of LF provides a strong antimicrobial potential by limiting the availability of Fe^3+^ required for bacterial growth. LF bacteriostatic function has demonstrated its effectiveness in vivo and in vitro against a wide range of both Gram-positive and Gram-negative bacteria [35,36,37,38]. In addition, LF shows bactericidal properties related to its interaction with the bacterial surface, inducing alterations in the osmotic function of membranes. As a result of the LF’s highly cationic charge, it can interact with negatively charged molecules on the bacterial surface (such as lipoteichoic acid in Gram-positive bacteria or lipopolysaccharides (LPS) in Gram-negative bacteria), inducing damage to the lipid bilayer of the microbial membrane. Lipid bilayer disruption increases the membrane permeability, damaging or even leading to bacterial death [39,40,41]. In this line, other crucial functions of LF in the defense against pathogens include the inhibition of biofilm production (biofilms as a crucial component of bacterial virulence) and the promotion of the actions of other natural antimicrobials (such as lysozyme) by reducing the negative charge on the bacterial surface during its interaction with LPS or lipoteichoic acid [42,43,44].

### 3.4. Antiviral and Antifungal Activity

The antiviral capacity of LF has been reported through extensive studies on a wide range of viral infections [45]. Binding to host cell surface glycosaminoglycans (especially heparin sulfate (HSPG)) has been postulated as the central mechanism underlying protection against viral infections, inhibiting the interaction between the virus and the host cell [46]. The benefit of LF in the prevention of viral infection and replication (mainly through the activation of interferon α/β) has been demonstrated in vitro for numerous viruses affecting humans, such as human immunodeficiency virus (HIV), respiratory syncytial virus, herpesvirus, cytomegalovirus, hepatitis C virus, and SARS-CoV-2 [47,48,49,50]; nevertheless, its clinical usefulness in this field is currently limited. Likewise, LF also protects against fungal infections [51,52,53]. The ability of LF to damage the cell membranes of pathogens, as well as its function as an iron scavenger, have been shown to play a decisive role in this context. 

### 3.5. Anticarcinogenic Properties

The interest in LF as an anticarcinogenic strategy is supported by its capacity to bind free iron, prevent the formation of toxic species, modulate cytokine secretion, regulate the immune–inflammatory response, participate in cell growth, promote apoptosis, and activate NK cells [54,55,56,57]. Previous studies have shown that iron chelators prevent the progression of estrogen-dependent neoplasms by limiting the bioavailability of free iron, which can damage cellular DNA and promote tumor cycling, linking the antitumor properties of LF to its action as an iron scavenger [58]. Moreover, LF inhibits the growth of cancer cells by blocking the tumoral circle, induces the expression of antitumoral cytokines (such as IL-18), and regulates NK cell cytotoxicity and CD8+ T-lymphocytes [59,60]. In this regard, exogenous treatment with LF has been effective in inhibiting tumor growth, but the mechanisms underlying this effect are still relatively unknown. 

## 4. Lactoferrin in the Ocular Surface

The ocular surface is an anatomical complex that includes the tear film, eyelids, cornea, and conjunctiva. Constant exposure to the environment makes the ocular surface a critical area for injury and infection, requiring an optimal defense system to maintain its integrity. 

### 4.1. Role of Lactoferrin in the Tear Fluid

LF is one of the most abundant proteins in tear fluid [61,62]. At this level, LF is locally synthesized by the lacrimal gland, as well as corneal and conjunctival epithelial cells [63,64]. The tear LF concentration is around 1.3 mg/mL [30,65] or 1.84 mg/mL [66], markedly higher than the plasma LF concentrations (1 μg/mL). 

Tear fluid is a vital component of the ocular surface defense system, containing essential elements involved in corneal and conjunctival epithelium protection. This biological fluid is produced, released, and delivered coordinately by the lacrimal functional unit (LFU). The LFU englobes the lacrimal glands, meibomian glands, conjunctival globet cells, corneal and conjunctival epithelial cells, eyelids, and the interconnecting innervation (beginning with the stimulation of the corneal nerve endings and progressing to the modulation of the lacrimal gland activity). Specifically, the neural complex underlying tear production and secretion consists of interacting networks with afferent and efferent innervation for the different members of the LFU (Figure 3A). The afferent pathways originate in the extensively innervated cornea via nerve impulses, traveling through the ophthalmic branch of the trigeminal nerve (V1) to the cerebral pons. In the mid-brain, synapses occur with parasympathetic neurons of the facial nerve (VIIth cranial nerve), targeted at the pterygopalatine ganglion. At this point, the efferent pathways project signals to lacrimal secretory glands and goblet conjunctival epithelial cells and promote tear secretion by releasing specific neurotransmitters [66]. As such, the proper activity of the LFU is critical to maintaining tear homeostasis.

In the composition of the tear fluid, lipids, mucins, water, proteins, electrolytes, salts, and metabolites intimately coexist and define the three main layers of the tear structure, from the outermost to the innermost: lipid, aqueous, and mucin layers (Figure 3B). The lipid layer is composed of a mixture of lipids secreted by the meibomian glands, which play a fundamental role in tear stabilization. It is a biphasic structure containing an inner layer of polar lipids in contact with the aqueous layer and a surface layer of non-polar lipids [67]. The aqueous layer is an intermediate layer in contact with the polar part of the lipidic layer and with the mucoaqueous layer. Its composition includes proteins, electrolytes, salts, metabolites, glucose, and oxygen, essential for nutrition, moisturization, and protection of the epithelial tissues. Regarding the mucin layer, it consists of mucins and water, providing hydration, wettability, and lubrication to the superficial cornea and conjunctiva epithelia. Membrane mucins are mainly secreted by the superficial epithelial cells of the cornea and conjunctiva [66,68]. 

The disruption of the proper LFU functionality and impairment of the tear composition leads to homeostasis breakdown in the tear film, resulting in altered LF concentrations at the ocular surface. In addition, although most tear proteins (such as LF) are locally produced by the lacrimal gland and annexes [63], they can also be transferred from the plasma [69]. In this line, the eventual states of inflammation may promote the transfer of proteins from the plasma due to the increased permeability of the conjunctival capillaries. Previous studies have reported changes in the tear LF levels under a variety of ocular or systemic conditions [30,70,71,72,73], as detailed in the following sections. These variations have been essentially related to inflammatory, immunologic, or oxidative events that occur during the pathophysiology of such diseases, making tear LF an interesting diagnostic and therapeutic biomarker.

### 4.2. Lactoferrin in Cornea and Conjunctiva

The anatomical, cellular, and molecular barriers that constitute the ocular surface also contribute to self-regulating the proper functioning and preservation of the defensive microenvironment, although the lacrimal gland is the main effector of the eye’s secretory immune system, playing a critical role in ocular defense [63,64]. In this regard, corneal and conjunctival cells have been shown to produce immunological components with anti-inflammatory and antibacterial activity (LF and cytokines) that play an active role in the specific and nonspecific immune defenses [74]. 

Epithelial cells act as the first line of defense, in part, by activating the innate immunity [75]. The innate immunity keeps the resistance of the intact corneal epithelium despite its continuous exposure to the microorganisms that reside, as normal bacterial flora, in the conjunctival sac and the eyelid margins. In many cases, factors that are usually proinflammatory to other cell types do not induce a defense response in corneal epithelial cells [76]. This statement supports the fact that the innate immunity in the corneal epithelium differs from the conventional innate immunity observed in other tissues, indicating a symbiotic relationship between the epithelium and the microbes inhabiting the ocular surface. Nevertheless, a disruption in the epithelial integrity or an overreaction as a defense against exogenous or endogenous antigens can induce the initiation and perpetuation of the inflammatory response. It is especially important in the transparent and avascular corneal epithelium, where the formation of scar tissue as an inflammatory response to an aggressor can lead to opacification and severe loss of vision.

As a remarkable contribution to maintain this immunoprivileged environment on the ocular surface, the epithelial cells produce detectable amounts of LF, with higher expression in the conjunctival epithelial tissue than in the cornea [77]. The capacity of the epithelial cells to generate LF was firstly demonstrated by Franklin, Kenyon, and Tomasi, who noted immunofluorescent LF staining along the apical border of the lacrimal acinar epithelial cells [78,79]. Similarly, the histological study on the accessory lacrimal tissue and the finding of secretory granules in normal conjunctival epithelium confirmed both as sources of tear LF production [80]. At the corneal level, Rageh et al. [81] noted LF gene expression in epithelial cells scraped from corneal tissues of human donors. These results were also consistent with the strong signals of punctate immunohistochemical staining for LF located in the nuclei and cytoplasm of epithelial cells over the human donor corneal epithelium [81]. The endogenous production of LF by the corneal and conjunctival epithelium was also studied in murine and bovine animal models [77,81]. Remarkably, it was found that these epithelia in bovines produce LF and that the LF promoter can control the expression of a reporter gene, demonstrating that these cells are a real source of LF and can be used in vitro to explore the regulation of LF expression [77].

The dysregulation of LF, or the other iron-binding proteins, directly impacts on the iron transport within the tissues [82]. Iron is necessary for many biological processes, but excessive amounts can be toxic, leading to oxidative stress and the synthesis of highly reactive radicals that cause tissue damage [83]. As an example, the reduced amounts of LF observed in the tear film of keratoconus (KC) patients contribute to the leakage and accumulation of iron in the corneal epithelium. This fact is clearly visible through the Fleischer ring, one of the characteristic biomicroscopic signs affecting corneas with moderate or advanced KC, which consists of iron accumulation in the corneal epithelial basement membrane, acting as a clear indicator of an altered iron metabolism [84]. 

Although little is known about the corneal iron metabolism, more mechanisms apart from the iron transporters (such as LF) are involved, including intracellular iron storage proteins (such as ferritin), iron regulatory mechanisms (IRE/IRP system), or iron-exporting proteins (such as ferroportin and ceruloplasmin). It has been found that corneal epithelial cells, in contrast to other cell types, contain nuclear ferritin rich in H-chains [85]. In other cells, this ferritin is only found under pathological conditions. It is hypothesized that corneal epithelial cells have developed a nuclear ferritin-based mechanism to protect their DNA against oxidative damage, which is important for a tissue constantly exposed to ultraviolet light [85]. In addition, the corneal epithelium has a specific nuclear translocation mechanism whereby ferritin is guided from the cytoplasm to the nucleus by a protein called ferritoid, identified as a component of a unique ferritoid–ferritin nuclear complex for DNA protection [86,87,88]. 

It is worth noting that the total amount of iron accumulated in the corneal tissue is not entirely responsible for mediating the oxidative damage, being rather a consequence of the excess of the labile iron fraction that is accessible for interactions with peroxides. The availability of intracellular labile iron plays a crucial role in the signaling mechanisms that determine the decision between survival or death in cells. The regulated cell death associated with excessive levels of labile iron is called ferroptosis. Targeting labile iron with iron-chelating drugs, such as LF, offers potential therapeutic opportunities. Therefore, in view of the imbalances that can be generated in response to ocular surface disorders and the favorable effect of the topical pretreatment with LF in mice and rat models [89,90], there emerges the necessity to develop biomedical applications of LF that contribute to maintaining its permanence at the tear and tissue levels.

## 5. Ocular Surface Conditions with Lactoferrin-Related Affectation

The ocular surface includes the outer layer of the cornea, the conjunctiva, the lacrimal and meibomian glands, the eyelid margin, and the tear film [91]. These components are interconnected through a continuous epithelium, as well as the vascular, nervous, endocrine, and immune systems [92]. It has unique properties, special physiological mechanisms (tear production and drainage), and a predisposition toward specific diseases. The ocular surface is a very important part of the eye due to its functional requirement for vision. Ocular surface diseases (OSD) include disorders that disturb the structures and functions of the cornea, conjunctiva, and the associated ocular surface gland network, affecting the stability of the tear film [93]. The causes of OSD are age, dry eye syndrome, meibomian gland dysfunction, and environmental and genetic factors, among others. Dry eye is the most prominent pathology involving the ocular surface; other less frequent but very relevant diseases are ocular infections and corneal ectasias such as KC. In all of these diseases, lower tear LF concentrations have been described in comparison with the tear LF concentrations found in the normal ocular surface status [30,94,95,96]. Throughout this section, we will update the characteristics of the main OSDs with LF-related involvement, paying special attention to the potential of LF as a promising diagnostic and therapeutic target.

Figure 4 summarizes the main LF-signaling pathways involved in OSDs.

### 5.1. Dry Eye and Lactoferrin

“Dry eye disease (DED) is a multifactorial disease of the ocular surface characterized by a loss of homeostasis of the tear film, and accompanied by ocular symptoms, in which tear film instability and hyperosmolarity, ocular surface inflammation and damage, and neurosensory abnormalities play etiological roles” [91]. DED is more common in women and increases with age (about 5–50% of the elderly show signs of eye dryness). Topical medications, the daily usage of antiglaucoma medications, and ophthalmic surgery are also responsible for OSD [97]. There is an association between DED and autoimmune diseases, especially Sjögren’s syndrome (SS), which is characterized by chronic inflammation of the salivary and lacrimal glands [98]. Neurodegenerative diseases show an increased incidence with advancing age (Parkinson’s and Alzheimer’s disease). In these patients, a lower blink rate and decreased corneal sensitivity influence the severity of DED symptoms [99].

Selenium-binding lactoferrin is taken into corneal epithelial cells by a receptor and prevents corneal damage in dry eye model animals [100].

A hallmark of dry eye disease is hyperosmolarity of the tear film, which damages the ocular surface and leads to a cascade of signaling events that release inflammatory mediators, decreasing the stability of the tear film [92]. DED is considered a chronic disease with periods of exacerbation and complications, ranging from mild to severe visual disturbances. In addition, there are associations with depression, sleep disorders, and migraine headaches [101]. 

Several studies analyzed tear proteins from normal subjects and DED patients; the results demonstrated a significant reduction of LF associated with dry eye. A tear protein panel in early dry eye, including 160 patients with early to mild DED, highlighted a decrease in LF (2.11 ± 0.74 mg/mL in normal subjects vs. 1.47 ± 0.76 mg/mL in patients with early dry eye) correlated with subjective symptoms [102,103]. Protein content in the tear film may also change with aging and dry eye, and a decrease in the LF and lysozyme levels was observed, leaving the ocular surface at a higher risk of infection [104]. Besides, a reduced tear LF level was correlated with the severity of epithelial lesions in patients with primary and secondary Sjögren’s syndrome [105]. Thus, altered LF concentrations might represent a potential diagnostic biomarker for DED. The results of several prospective case–control studies with Sjögren’s syndrome patients [98] and DED patients [106], as well as a meta-analysis study in human tears and ocular diseases [107], suggest that the LF level in tears is a good candidate as a DED diagnostic biomarker. Proof of this is the rapid, portable test by microfluidic technology (TearScan 300 MicroAssay System) designed to measure lactoferrin levels in human tear fluid for improving the diagnosis of Sjögren´s syndrome and other forms of DED diagnoses [95]. 

The topical application of LF has shown to reduce corneal epithelial damage in mice models, and to promote corneal wound healing after alkali burn injury [89]. The rationale for the use of LF derives from its capacity to address the vicious cycle of dry eye disease, especially the underlying inflammation and oxidative stress [23,108]. In a study, patients with Sjögren’s syndrome supplemented with oral LF showed tear film stability and ameliorated dry eye symptoms [105]. Another study reported its efficacy in improving ocular surface parameters in DED patients induced by cataract surgery [109]. Therefore, specific topic treatments with LF can attenuate pathological changes and initiate tissue repair processes in DED [94,104].

### 5.2. Ocular Surface Infections and Lactoferrin

Corneal opacity represents a cause of blindness, and any significant aggression to the cornea, such as an infection, can result in corneal opacity with visual impairment. Infectious keratitis (IK) has shown to be the most common cause of corneal blindness in developed and developing countries, with an estimated incidence ranging from 2.5 to 799 per 100,000 population, affecting individuals across all age groups [110]. IK can be caused by bacteria, fungi, viruses, parasites, and polymicrobial infections. Bacteria and fungi have shown to be the most common microorganisms for corneal infection in the world´s population. Viral and *Acanthamoeba* spp. keratitis represents important causes in developed countries [111]. Trauma, contact lens wear, ocular surface diseases, ocular post-surgery, topical steroids, and systemic immunosuppression have been shown to be the major risk factors for IK.

Supporting the LF involvement, Comerie-Smith et al. [96] observed markedly decreased levels of LF in tears of asymptomatic HIV-positive patients (85.8 mgs/dcl of LF, compared to 156 mgs/dcl in HIV-negative patients), and the increased colony counts of bacterial flora in their lids (4.1 colonies/patient, compared to 1.5 colonies/patients in the control group).

The activity of LF against microbial invasion was the first biological property to be discovered and the most studied one. LF is able to damage the bacterial membrane and prevent neutrophil inflow, downregulating the inflammatory response [112]. LF has shown to be efficient against the ingrowth of several bacteria, including *Hemophilus influenza*, *Escherichia coli*, *Streptococcus* spp., *Staphylococcus* spp., and *Pseudomonas* spp. [113]. It has been reported that LF could act against Pseudomonas aeruginosa by inhibiting colonization and biofilm formation on the contact lens surface [114]. Fungal keratitis is most commonly caused by *Aspergillus* spp. and *Fusarium* spp. [115], and LF was also able to prevent biofilm formation over contact lenses.

LF, as a multifunctional protein, also exhibits efficacy in the setting of viral infectious processes. Several studies have highlighted the activity of LF against cytomegalovirus (CMV), herpes simplex virus (HSV), hepatitis C virus (HCV), human immunodeficiency virus (HIV), parainfluenza virus (PIV), human papillomavirus (HPV), and adenovirus [50]. The antiviral activity of LF lies in the early phase of infection, preventing the virus entry into host cells. This explains the great efficacy of LF against HSV infection and preventing HCV and HIV entry into host cells [116,117]. In addition, the immunomodulatory effects of LF may be related to its antiviral action. Similar mechanisms have been proposed for SARS-CoV-2 (COVID-19) conjunctivitis infection, in which LF has demonstrated the ability to inhibit viral replication in vitro [118]. Finally, a study showed the amoebicidal effect of LF against *Acanthamoeba* spp. clinical-isolated trophozoites; these results might lead to the prevention of contamination by *Acanthamoeba* spp. in contact lens case stocks [119]. 

The thorough treatment of ocular surface infection is not limited to eliminating the microorganism; it also involves the prevention of ocular scarring and opacity, the minimization of tissue damage, and the preservation of visual acuity [120]. In this regard, topical LF answers the need to develop new treatments to target infections at the ocular surface.

### 5.3. Keratoconus and Lactoferrin

KC is a progressive and bilateral ecstatic disorder characterized by weakening, thinning, protrusion, and chronic degeneration of the corneal tissue [121]. KC affects young adults, causes severe visual disability, and leads to a great impact on the quality of life [122]. Its prevalence ranges from 0.0002% to 2.34% [123], presenting a very rapid progression in its initial stages. Nowadays, there are no approved treatments to prevent its development; surgical treatments dedicated to the replacement or regularization of the corneal surface provide limited visual recovery, and corneal collagen crosslinking (CXL), which is the only conservative treatment that has been shown to be effective in slowing down its progression, is not successful in all cases [124]. Consequently, the need arises in finding pharmacological treatments that contribute to restoring the molecular imbalances which trigger the disease. In this line, LF has been shown as a crucial molecule in the pathogenesis of KC.

Numerous research studies have recognized the role of oxidative stress, innate immunity, and inflammation pathways in the KC pathogenesis [125,126]. Higher levels of damaging free radicals and lower levels of antioxidant molecules were reported in KC samples [127]. An overexpression of TLR2 and TLR4 was observed in correlation with increased inflammatory mediators and NF-kB factors and also interrelated with the severity of the disease [128,129,130]. Furthermore, a gradual increase in inflammatory mediators comprising cytokines, cell adhesion molecules, and matrix metalloproteases (such as IL-6, TNFα, and matrix metalloprotease 9 (MMP-9)), was described as an important contributor to KC progression [125,131]. 

In addition, iron overload is a hallmark clinically observable through the Fleischer’s ring by a slit-lamp study in corneas with moderate to advanced KC. Fleischer’s ring consists of iron deposits in the epithelial basement membrane, surrounding the base of the cone. This altered iron metabolism may be enhanced by the lower tear and serum LF concentrations observed in KC patients (tear LF levels are 1.54 times significantly lower in KC than in controls, and serum LF levels are 2.6 times significantly lower in KC than in controls), which strongly correlate with the disease’s immune, inflammatory, and clinical status [30,132,133]. The reduced concentrations of LF, together with the decrease of other iron-binding proteins like serotransferrin, may contribute to iron filtration and accumulation in the corneal epithelium, leading to an oxidative microenvironment and cell damage or cell death processes such as ferroptosis [72]. Based on these findings, as well as on the known biological multifunctionality of LF, it is obvious to hypothesize that restored LF levels on the tear film of KC patients may trigger the modulation of oxidative, immune, and inflammatory factors involved in KC. In addition, iron-chelating proteins, such as LF, have been shown to successfully prevent ferroptosis-related cell death.

Figure 5 shows an overview of the ocular surface conditions related to LF, suggested beneficial contributions that LF could provide to their pathology.

To date, several preclinical studies have proposed biocompatible LF delivery systems for topical ophthalmic administration, in order to accomplish a new pharmacological alternative to invasive surgical procedures for the treatment of KC [134,135,136]. These studies were developed with the aim of increasing the permanence of LF on the ocular surface and improving the penetration of LF into the cornea. In this line, Pastori et al. [137] suggested that LF-loaded contact lenses could represent a promising new therapeutic strategy to treat ocular surface conditions. 

## 6. Drug Delivery Alternatives for the LF Topical Administration

The eye is one of the best-isolated organs, as it has effective protection mechanisms against external agents and exogenous substances clearance. The average volume of the precorneal tear film is 7–10 μL. This volume may increase at 30 μL after the instillation of the eye drop, decreasing again to 7 μL with the first blink, causing an approximate decrease in the concentration of the instilled drug by 80%. Moreover, it is estimated that the renewal of the tear film is 1.2 μL per minute, which suggests that hardly any drug remains on the ocular surface five minutes after the instillation of the eye drops. Thus, the permanence and bioavailability of drugs administered by conventional eye drops on the eye surface are very low. Typically, less than 5% of the administered dose by classic eye drops can be absorbed by ocular tissues [138,139]. 

In the last decades, different strategies have been proposed to ameliorate topical ophthalmic therapeutic effects and minimize the non-desirable adverse effects. Different alternatives based on the improvement of the drug permanence on the ocular surface, drug bioavailability, and penetration into the ocular tissues have been proposed [140]. Several hydrogel-based ophthalmic formulations, as well as different hydrophobic colloidal systems, have been recently reported as alternatives to improve the drug permanence on the ocular surface, drug absorption into the cornea and controlled drug release [138,139]. 

LF has demonstrated an active contribution, not only on the ocular surface but also in the corneal epithelium and stroma. Considering the high molecular weight of LF (80 kDa), its corneal mean residence time (MRT) and its diffusion ability into the epithelium are essential parameters to be considered to achieve significant LF concentrations. According to Subrizi et al. [141] large molecules with molecular weights up to 5 kDa can permeate across the conjunctiva and macromolecules as LF can permeate across the sclera. Consequently, if it is necessary to improve LF uptake in the corneal epithelium and stroma to increase its efficacy, LF delivery systems that promote an increase in LF residence time on the ocular surface may be a good alternative.

Despite the availability of several systems concerning the release of LF for different uses previously described in the literature, in the specific case of topical ophthalmic administration cyclodextrin, nanotechnology, and mucoadhesion have been proposed as the main technological approaches (Figure 6) [138,141]. Different researchers have studied polysaccharide nanoparticles, biodegradable polymeric nanoparticles, lipid systems based on liposomes, and nanostructured lipid systems or medicate soft contact lens to increase the LF ocular permanence and diffusion. Nonetheless, there are hardly any studies of its behavior on the ocular surface and cornea at present. 

In this line, Varela et al., in 2021 [134], proposed the preparation of two different types of LF-loaded chitosan mucoadhesive nanospheres, crosslinked with sodium tripolyphosphate (TPP) and with sulfobutylether-β-cyclodextrin (SBE-β-CD), respectively, for the KC treatment. Taking advantage of the presence of sialic acid and negative charges in the ocular mucosa, they proposed nanoparticles with a positive surface charge to promote mucoadhesive interactions. The researchers developed both types of nanospheres using an ionotropic gelation technique. With this methodology, the researchers obtained nanoparticles with a mean diameter of less than 300 nm, ζ-potential values ranging from +17.13 to +19.89 mV, and an LF immobilization yield of around 50%. The polysaccharide nanoparticles showed excellent stability over a three-month storage period under biological conditions of pH and ionic strength, as well as an LF-controlled release for more than 24 h in an artificial tear medium. The in vitro, ex vivo, and in vivo bioadhesion studies showed good mucoadhesive properties of both types of nanospheres with a prolonged residence time on the ocular surface. The in vivo ocular surface permanence studies developed by means of a computerized PET/CT (Positron Emission Tomography/Computerized Tomography) image analysis showed t_1/2_ and MRT values in the cornea of 114 min and 127.3 min for CS/TPP, and 60.5 and 89.9 min for CS/SBE-β-CD, respectively, vs. the 17.7 min and 59.1 min of the free radiolabeling molecule. Both nanospheres exhibited higher ocular permanence compared to the control, with apparently low permanence of CS/SBE-β-CD vs. CS/TPP. The authors suggested that the increased ionic strength and osmolarity values caused by the high sodium concentration of the SBE-β-CD derivative may lead to the aggregation of the CS/SBE-β-CD nanoparticles, and an increase in the blinking after instillation may reduce its surface ocular permanence.

Recently, López-Machado et al. (2021) [142] and Varela et al. (2022) [135] have also proposed the use of biodegradable polymeric nanoparticles to increase the concentration of LF in the ocular tissues improving the ocular surface permanence time. Varela et al. [135] developed non-toxic LF-loaded nanospheres and nanocapsules for KC treatment by a one-step and a two-step nanoprecipitation method, respectively, using a variety of 50:50 polyacrylic-polyglycolic acid (PLGA) copolymer of different molecular weights (10,000, 17,000, 24,000, and 38,000 Da). All nanospheres showed an average diameter between 100 and 150 nm regardless of the PLGA composition. Nanocapsules were larger in size, with diameters between 150 and 300 nm, depending on the molecular weight of the PLGA. In contrast, the ζ-potential values were slightly higher in the nanospheres than in the nanocapsules. The authors also demonstrated the low loading capacity of the nanospheres compared to the nanopcapsules to immobilize. Nanospheres showed an LF loading capacity of less than 10% in comparison to the nanocapsules obtained with the lower molecular weight PLGA, with dosage values of 60%, as well as production yield and encapsulation efficiency values above 80%. PLGA nanocapsules were stable during storage under biological conditions of pH and ionic strength. The in vitro release studies showed a LF-controlled release of almost 24 h with a release kinetic dependent on the molecular weight and the PLGA variety used in the manufacture of the nanocapsules. The in vivo ocular permanence studies by computerized PET/CT showed a t_1/2_ higher than the reference (protein solution), with values of 93.31 min and 51.32 min for PLGA nanospheres and nanocapsules, respectively. The t_1/2_ values were in the same order as those obtained in previous studies with chitosan nanospheres. 

López-Machado et al. [142] prepared PLGA nanocapsules (50:50; 38,000 Da molecular weight) for the treatment of inflammatory processes of the anterior segment of the eye. The researchers used a central composite experimental design to obtain the best composition and production conditions. Optimized nanoparticles showed a monodisperse population in terms of size, with an average diameter below 250 nm and high negative ζ-potential values. In vitro cellular studies showed that the nanoparticles were not cytotoxic, had the ability to incorporate through the LRP1 pathway, and were able to inhibit the LPS-induced inflammatory response in an HCE-2 cell line. Ex vivo permeation studies using isolated corneas from New Zealand rabbits show that LF immobilized into nanoparticles can permeate slightly faster than free LF through the cornea. The authors found significant differences in the values of flux, permeability, and amount permeated at 24 h between LF-loaded nanocapsules and free LF. However, no differences were found in the amount of LF retained in the cornea at the end of the experiment. In vivo ocular studies of tolerance (Draize test) and anti-inflammatory efficacy carried out in a rabbit model pointed to the ability of the nanoparticles to prevent and treat ocular inflammation. The authors used a model of inflammation induced by arachidonic acid and studied the protective and curative effects of the LF-loaded PLGA nanoparticles, finding effectiveness in both treatments. 

Varela et al. (2022) [136] also proposed LF-loaded nanostructured lipid carriers (NLC) to improve LF bioavailability for the treatment of KC. The NLC were prepared by a double emulsion/solvent evaporation technique with a thermosensitive gel core to enhance the LF immobilization into the lipid formulation. The NLCs were monodisperse with an average diameter of 120 nm. The encapsulation efficiency and loading capacity of NLC were high depending on the LF concentration employed in the elaboration procedure. The process was improved when the LF concentration in the manufacturing medium was higher than 1 mg/mL. At this concentration, the encapsulation efficiency and loading capacity were 80% and 70%, respectively. The NLCs proved to be non-toxic and stable in storage under biological conditions. Moreover, the in vivo ocular permanence studies by the PET/CT technique showed NLC with excellent mucoadhesive properties, with values of 107.82 and 141.33 min for t_1/2_ and MRT, respectively. 

Lopez Machado et al. (2021) [143] also proposed the use of hyaluronic acid-coated liposomes as a new lipid formulation for the treatment of dry eye disease and ocular inflammation. The authors produced fat-free soybean phospholipid liposomes with phosphatidylcholine and cholesterol by the lipid film hydration method combined with a high-pressure homogenization process. Hyaluronic acid-coated liposomes showed an average size of 90.5 nm, a positive surface charge with ζ-potential values of +20.5 mV, and an encapsulation efficiency of 50%. In vitro and ex vivo corneal permeation assays (performed on isolated New Zealand rabbit corneas) showed an effective control of the release and an improvement of LF corneal permeation from liposomes. Ex vivo permeation studies showed higher flux values, permeability, and amount permeated at 24 h for LF-loaded liposomes compared to a free LF solution, as well as a decrease in the amount retained in the cornea when liposomes were used. Comparing these results with those obtained in the previous study with nanoparticles [142] an increase in the values of all pharmacokinetic parameters for both free LF and immobilized LF into liposomes, and a reduction in the amount of LF retained in the cornea in the case of liposomes were observed. Two different in vivo models using benzalkonium chloride or LPS treatment were developed to study the efficacy of liposomes in the treatment of DED and ocular inflammation. Hyaluronic acid-coated liposomes showed good activity in the Schirmer test in the DED animal model and an improvement in the prevention and treatment of ocular inflammation compared to the control, as well as to the group treated with a solution of free LF, was proved.

Marketed soft contact lens loaded with LF has also been used for the glycoprotein release in the eye for protection against oxidative stress [137] and to counteract cytotoxicity caused by keratoconic process [144]. Initially, Pastori et al. [137] investigated the ability of three types of commercial contact lenses to load and release LF: the silicone-based hydrogel filcon V and filcon IB, and the hydroxyethyl methacrylate-based hydrogel galyfilcon A. Filcon V showed better LF loading and releases behavior with a loaded level of 61 μg of glycoprotein per lens. LF released from the lens maintains its antioxidant activity in human epithelial cell culture, showing a protective effect against oxidative stress. The same authors study the antioxidant activity of LF loaded in the silicone-based hydrogel filcon V on the epithelial cells incubated with keratoconic tears. The incubation of epithelial cells with tears of KC patients produces an increase in cell mortality compared with the incubation with tears of healthy patients. Both works show the in vitro efficacy of LF-loaded contact lenses for protection against oxidative stress.

Finally, LF has also been proposed to treat pathologies of the posterior segment of the eye. An example is the work of Ahmed et al. in 2014 [145] that proposed the use of LF-coated nanoparticles as a new strategy to improve cellular uptake in the retinoblastoma treatment.

Therefore, efforts are being made to achieve effective LF delivery systems to the ocular surface for the treatment of inflammatory and degenerative diseases of the eye. Pharmaceutical nanotechnology has demonstrated its efficacy to interact with the ocular mucosa and epithelia, increasing the permanence of the glycoprotein on the ocular surface. The nanoparticles developed have shown good in vitro and in vivo efficacy in animal models in the treatment of inflammatory pathologies or with an important oxidative component. Consequently, these works have laid the groundwork for the future clinical use of LF delivery systems with applicability not only in KC but also in other ocular surface conditions with LF involvement.

To our knowledge, there are no topical ophthalmic LF products on the market nor clinical studies of topical ophthalmic LF formulations. Only a clinical study in patients with dry eye induced by cataract surgery [109] and a small clinical study conducted on 10 patients with Sjögren’s Syndrome [104] but where LF was administered orally has been developed.

An additional problem in the development of new formulations of Lf with the potential to be commercialized is their stability. There are several studies that determine the stability of LF in different media and delivery systems [146,147,148]. Kim and coworkers studied the stability of LF in solution in the presence of various excipients [143]. They found that arginine and polysorbate 80 can protect the molecule from physical or chemical destabilization. After the addition of arginine, polysorbate 80, and trehalose, it is possible to obtain a solution that remains stable for more than 150 days in the refrigerator. However, stability declines at room temperature or above.

Yao et al. studied the physicochemical stability of LF-loaded liposomes and solid lipid nanoparticles modified by a chitosan or pectin [147]. All the LF-loaded liposomes show a rapid release of the LF overall at 40 °C and a complete degradation after 180 days of storage time, whereas almost 30% of intact Lf still remained after 180 days in solid lipid nanoparticles. However, the instability processes in lipid formulation are related to the premature release and expulsion of LF to the environment than to its degradation within the lipid systems. Finally, there is a complete review of the effects of technological treatments used in the treatment and preservation of food on LF [148]. It can be seen how thermal treatments at high temperatures produce the aggregation and denaturation of LF, as well as some high-pressure homogenization processes when very high-pressure values are used. Other processes such as spray-drying or freeze-drying respect the LF structure. 

CDs can be a candidate to contribute to the stabilization of the LF in the nanomedicine and hydrogels. CDs are cyclic oligosaccharides that show a good ability to form complexes with drug molecules and to improve their physicochemical properties without molecular modifications, via drug/host interaction [149,150]. The capacity of CDs to interact with proteins is well known [151,152,153,154,155]. Different mechanisms have been described by which cyclodextrins interact with proteins improving physical and chemical stability [154]. CDs can form inclusion complexes with amino acids, mainly βCD derivatives and hydrophobic and aromatic residues of Phe, Tyr, His, and Trp, modulating the solvent exposure to hydrophobic amino-acidic residues [151,152,153,154]. Additionally, the surface activity of some CD-derivative can contribute to protein stabilization by reducing the protein surface adsorption [154]. DCs can prevent protein aggregation and adsorption through these mechanisms and improve stability against proteases. Consequently, CDs derivatives are excellent candidates for improving the chemical and physicochemical stability of proteins in the solid and liquid states [155].

In recent years, 3D printing has emerged as a promising technology for creating complex structures and materials with precise control over their physical properties [156]. There has been some research into 3D printing lactoferrin, particularly for its potential use in biomedical applications but not specifically for ophthalmic applications. One example is the 3D printing of lactoferrin-loaded alginate hydrogel scaffolds using a bioprinter [157]. The researchers found that the printed scaffolds had good biocompatibility and could support the growth of human mesenchymal stem cells. Another example is the development of a 3D-printed lactoferrin-based hydrogel that could be used as a wound dressing [158]. The hydrogel had good mechanical properties and could release lactoferrin in a controlled manner. Overall, 3D printing lactoferrin holds promise for the development of new therapeutic and biomedical applications. However, further research is needed to fully understand this technology’s potential and optimize the printing process of lactoferrin-based hydrogels for ophthalmic applications.

## 7. Concluding Remarks and Future Directions

Through this review, it has been possible to elucidate the potential of LF as a promising therapeutic target for a wide range of ocular surface conditions. Based on the LF multifunctional pattern, which includes its anti-inflammatory, immunomodulatory, antioxidant, antiviral, antibacterial, and antitumoral properties, the LF topical application may provide highly beneficial contributions to dry eye, ocular infections, and KC disease, among others. In this regard, some of the most remarkable pharmacological contributions of LF are its capacity to reduce corneal epithelial damage by promoting wound healing, increase the tear film stability and attenuate dry eye symptoms, cope with infectious insult, immunomodulate the inflammatory response, and restore the iron homeostasis in conditions of iron dysregulation, as happens in KC. In comparison with the current therapeutic approaches available for the mentioned OSDs (for example, antibiotics for eye infections and artificial tear therapy for DED, etc.), the application of LF would provide a wide spectrum of protection and could be used to support and reinforce existing treatments. The clinical outcome of this approach is mainly based on reconstituting and reinforcing LF levels to maintain the natural state of the ocular surface, taking advantage of the benefits it provides.

At present, several studies are underway on the development of effective LF treatments, but most of them are limited to the preparation of LF delivery systems and the subsequent characterization of its delivery and passage across the corneal tissue. In this way, the improvement of the drug permanence on the ocular surface, drug bioavailability, and penetration into the tissues are quite a challenge to enhance the efficacy of the pharmacological treatments. With this purpose, in vitro studies have been carried out to develop different drug delivery strategies for LF administration on the ocular surface, and in vivo studies of the LF ocular permanence and efficacy have been performed on healthy animals or inflammation/dry eye models. However, in KC disease, the lack of consistent animal models hampers the study of the efficacy of these types of drug delivery systems. Additionally, although ex vivo corneal permeation studies have been developed, there are still no in vivo eye pharmacokinetic or tissue distribution studies of topically administered LF. It will be therefore necessary to develop such studies to confirm the use and effectiveness of new formulations.

The management of KC with LF is expected to be a chronic treatment, as will be the case for other ocular surface conditions. Nowadays, the biocompatibility and ocular toxicity of the LF administration have only been studied for short periods, a weakness in determining the ocular surface response to LF long-term treatment. Therefore, more studies are necessary to highlight the behavior of these LF strategies on the human ocular surface.

Numerous clinical trials are currently ongoing for the use of LF in the fields of cancer, intestinal diseases, and COVID-19. Nevertheless, in the ophthalmology field, only one randomized controlled trial about the effect on DED of a combined dietary supplement (for oral administration) containing lactoferrin (among other components, such as fish oil, zinc, vitamin C, lutein, vitamin E, γ-aminobutanoic acid, and Enterococcus faecium WB2000) has been reported [159]. Nowadays, there are no clinical trials in progress focused on LF treatment via ophthalmic administration in eyes affected by any of the OSDs mentioned in this review.

The current challenges for the proposed field include the creation of a biocompatible topical ocular formulation that allows adequate permanence on the ocular surface and that can even be compatible with existing treatments.

In summary, the motivation of this review was to highlight the relevance of LF and its biomedical applications on the ocular surface. More concretely, the uniqueness of this revision in comparison with previous ones includes: the latest update on the topic, the wide scope of LF-related OSDs, and the most recent information about the delivery systems for LF ocular topical application, as well as an easily visualized information through well-crafted illustrations on the LF signaling pathways and the LF beneficial contributions in the mentioned OSDs.

## Figures and Tables

**Figure 1 pharmaceutics-15-00865-f001:**
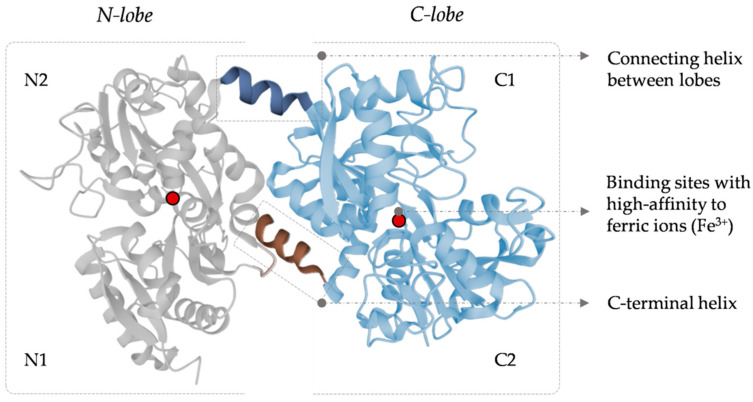
Human LF secondary and tertiary structures. Self-created figure using Uniprot elements (free access at https://www.uniprot.org (accessed on 27 February 2023)).

**Figure 2 pharmaceutics-15-00865-f002:**
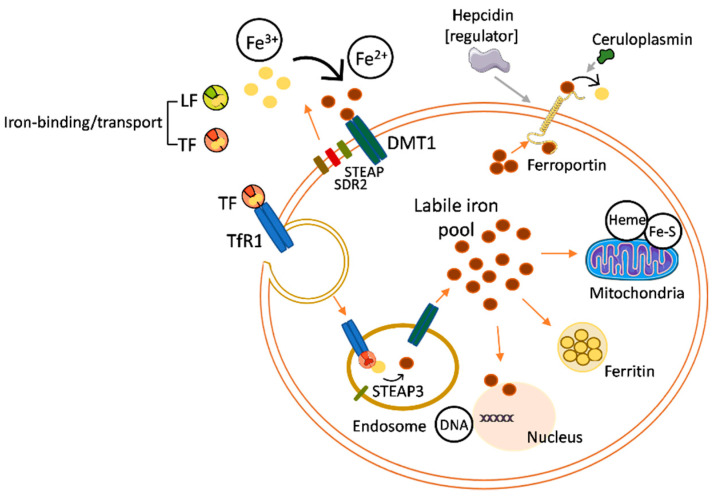
Summary of the iron regulation mechanisms within the cell. Abbreviations: DMT1, divalent metal transporter 1; DNA, deoxyribonucleic acid; Fe^2+^, ferrous ion; Fe^3+^, ferric ion; Fe-S, iron sulphide; LF, Lactoferrin; SDR2, stromal cell-derived receptor 2; STEAP, six transmembrane epithelial antigens of the prostate family; STEAP3, STEAP family member 3 metalloreductase; TF, transferrin; TfR1, transferrin receptor 1. Self-created figure using elements with Creative Common (CC) license.

**Figure 3 pharmaceutics-15-00865-f003:**
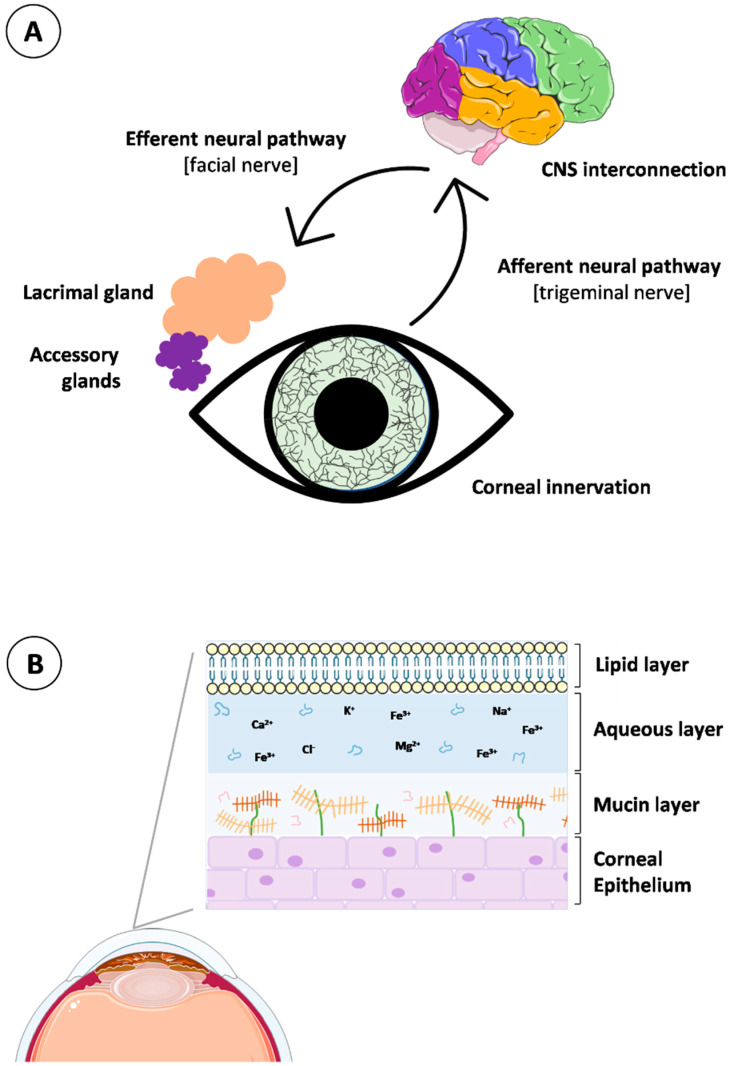
Lacrimal functional unit and tear fluid: (**A**) the neural complex and (**B**) the tear composition. Self-created figure using elements with Creative Common (CC) license.

**Figure 4 pharmaceutics-15-00865-f004:**
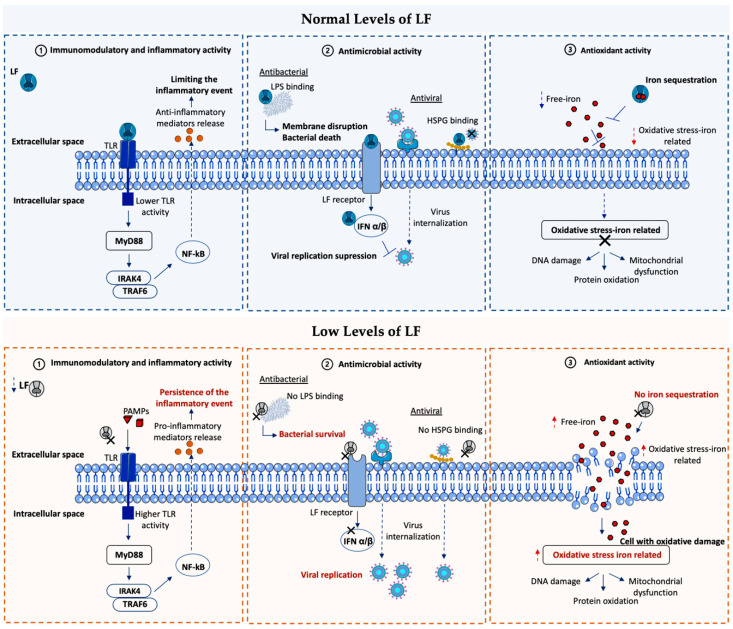
LF-signaling pathways involved in mechanisms leading to major OSDs. (1) About OSDs associated with immune–inflammatory events (such as keratoconus or dry eye), LF modulates the immune–inflammatory response through Toll-like receptors (TLRs), competing for the binding site with pathogen-associated molecular patterns (PAMPs) and promoting the release of inflammatory mediators and the resolution of the inflammatory event. (2) In terms of ocular surface infections, binding to bacterial LPS and cell surface HSPGs, as well as the activation of IFN, represents the key mechanisms for protection against bacteria and viruses. (3) Regarding the OSDs related to high levels of oxidative stress (such as keratoconus or dry eye), the antioxidant power of LF due to its capacity as an iron scavenger limits the amount of free or unbound iron in the extracellular space, preventing the activation of harmful oxidative reactions. Self-created figure using elements with Creative Common (CC) license.

**Figure 5 pharmaceutics-15-00865-f005:**
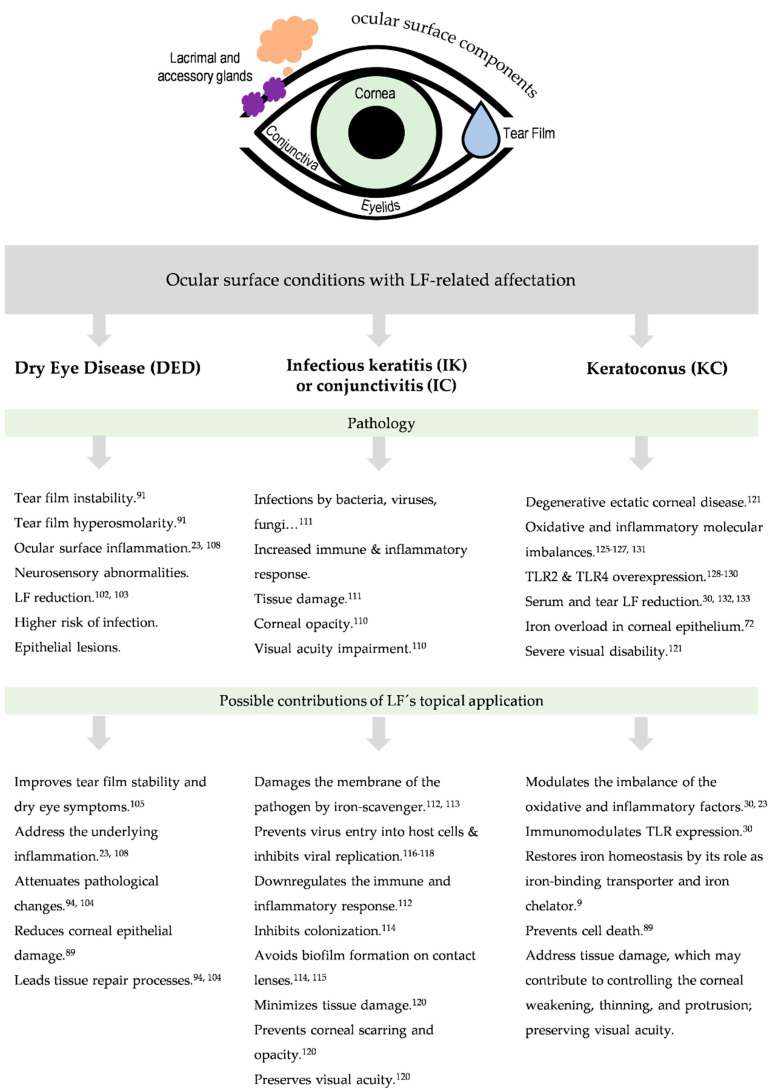
Schematic summary of the main ocular surface conditions with LF-related affectation, along with the potential beneficial contributions that LF could make to their pathology. Self-created figure using elements with Creative Common (CC) license. The superscript numbers belong to the references cited in this manuscript [9,23,30,72,89,91,94,102,103,104,105,108,110,111,112,113,114,115,116,117,118,120,121,125,126,127,128,129,130,131,132,133].

**Figure 6 pharmaceutics-15-00865-f006:**
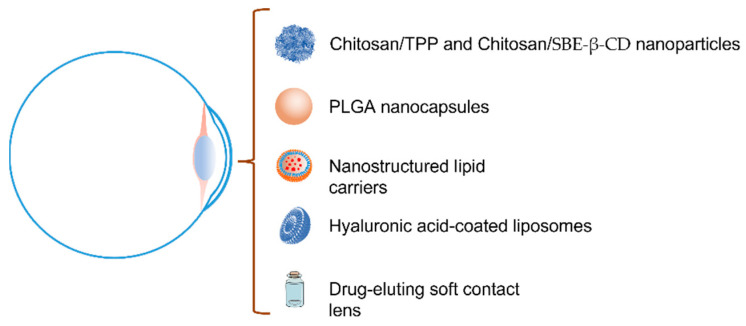
Different technologies used to improve the ocular drug bioavailability of lactoferrin.

## Data Availability

Not applicable.

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
