# Peer review of "Biomedical Applications of Lactoferrin on the Ocular Surface"

_pharmaceutics, 2023, doi:10.3390/pharmaceutics15030865_

Round 1

Reviewer 1 Report

This manuscript summarizes the function of lactoferrin on ocular disorders as well as its topical administration towards clinical applications. I am listing some concerns.

(1)     Lactoferrin concentration in disease status should be clearly illustrated (in term of in tear fluid, cell, or tissue environment).

(2)     The signaling pathway of lactoferrin (in form of figure or scheme) in ocular disease should also be summarized.

(3)     The clinical outcome of applying lactoferrin should be explained in comparson with current therapeutic approach (for example, in terms of DED, artificial tear therapy should be applied as a control to indicate the clinical potential of applying lactoferrin).

(4)     For Figure 4, it might be better to illustrate the drug delivery approach for lactoferrin only.

(5)     The bioavailability of lactoferrin (or advantages and weakness) in terms of different delivery approach should be illustrated.

(6)     Is there any clinical trial on lactoferrin therapy? If so, it might also be valuable to show these information.

Author Response

Reviewer 1 report:

This manuscript summarizes the function of lactoferrin on ocular disorders as well as its topical administration towards clinical applications. Reviewer 1 listed some concerns:

(1)   Lactoferrin concentration in disease status should be clearly illustrated (in term of in tear fluid, cell, or tissue environment).

As recommended by the reviewer, the concentrations of lactoferrin in the different pathological conditions mentioned have been clarified. In this regard, the following information has been included:

  • Lines 343-345: “In all of these diseases, lower tear LF concentrations have been described in comparison with the tear LF concentrations found in normal ocular surface status [30, 94–96].”
  • Lines 383-384, concentrations of LF in the dry eye disease: “(2.11±0.74 mg/ml in normal subjects vs. 1.47±0.76 mg/ml in patients with early dry eye)”.
  • Lines 417-421, concentrations of LF in ocular surface infections: “Supporting the LF involvement, Comerie-Smith et al [96] observed markedly decreased levels of LF in tears of asymptomatic HIV-positive patients (85.8 mgs/dcl of LF, compared to 156 mgs/dcl in HIV-negative patients), and the increased colony counts of bacterial flora in their lids (4.1 colonies/patient, compared to 1.5 colonies/patients in the control group)”.
  • Lines 473-474, concentrations of LF in the keratoconus disease: “(tear LF levels are 1.54 times significantly lower in KC than in controls, and serum LF levels are 2.6 times significantly lower in KC than in controls)”.

(2)  The signaling pathway of lactoferrin (in form of figure or scheme) in ocular disease should also be summarized.

 We appreciate the reviewer’s suggestion. As a result, a representative figure of the LF signaling pathways involved in ocular diseases was included in Lines 350-361 of the revised manuscript. [Figure 4]

(3) The clinical outcome of applying lactoferrin should be explained in comparison with current therapeutic approach (for example, in terms of DED, artificial tear therapy should be applied as a control to indicate the clinical potential of applying lactoferrin).

In order to clarify the outcome of lactoferrin application in ocular surface diseases compared to current therapeutic approaches, the following information was added to the Concluding Remarks and Future Directions of the revised manuscript:

  • Lines 707-713: “In comparison with the current therapeutic approaches available for the mentioned OSDs (for example, antibiotics for eye infections and artificial tear therapy for DED…), the application of LF would provide a wide spectrum of protection and could be used to support and reinforce existing treatments. The clinical outcome of this approach is mainly based on reconstituting and reinforcing LF levels to maintain the natural state of the ocular surface, taking advantage of the benefits it provides.

 (4) For Figure 4, it might be better to illustrate the drug delivery approach for lactoferrin only.

 The authors thank the reviewer for the excellent suggestion. Figure 4 [Figure 6 in the revised manuscript, Line 519] has been modified to appropriately illustrate the drug delivery approach for LF.

 (5) The bioavailability of lactoferrin (or advantages and weakness) in terms of different delivery approach should be illustrated.

 The authors acknowledge the reviewer for pointing this out. As far as is known so far, there are no pharmacokinetic or ocular bioavailability studies performed for LF formulations. The reviewed articles show that ex vivo corneal permeation studies, in vivo drug clearance from the ocular surface and in vivo efficacy studies using inflammation models have been performed for the lactoferrin-loaded developed systems. Nevertheless, the authors have not found information on pharmacokinetic or tissue distribution studies in the eye after topical administration of LF formulations. Despite this, the authors have submitted a revised version of the manuscript, addressing the concerns/doubts raised. New information was incorporated or modified in this new version, as follows:

  • Lines 590-595: “Ex vivo permeation studies using isolated corneas from New Zealand rabbits show that LF immobilized into nanoparticles can permeate slightly faster than free LF through the cornea. The authors found significant differences in the values of flux, permeability, and amount permeated at 24 hours between LF-loaded nanocapsules and free LF. However, no differences were found in the amount of LF retained in the cornea at the end of the experiment.”
  • Lines 620-627: “Ex vivo permeation studies showed higher flux values, permeability and amount permeated at 24 hours for LF-loaded liposomes compared to a free LF solution, as well as a decrease in the amount retained in the cornea when liposomes were used. Comparing these results with those obtained in the previous study with nanoparticles [130], an increase in the values of all pharmacokinetic parameters for both free LF and immobilized LF into liposomes, and a reduction in the amount of LF retained in the cornea in the case of liposomes were observed.”
  • Lines 724-727: “Additionally, although ex vivo corneal permeation studies have been developed, there are still no in vivo eye pharmacokinetic or tissue distribution studies of topically administered LF. It will be therefore necessary to develop such studies to confirm the use and effectiveness of new formulations.”

 (6) Is there any clinical trial on lactoferrin therapy? If so, it might also be valuable to show these information.

 Numerous clinical trials are currently ongoing for the use of LF in the fields of cancer, intestinal diseases, and covid-19. Nevertheless, in the ophthalmology field, only one randomized controlled trial about the effect on DED of a combined dietary supplement (for oral administration) containing lactoferrin has been reported. The authors have performed an exhaustive search for information and have not found any details about topical ophthalmic LF products on the market or clinical studies of topical ophthalmic LF formulations. In the reviewed bibliography, the authors have also noticed the existence of a clinical study in patients with dry eye induced by cataract surgery, as well as a small study in Japan conducted in 10 patients with Sjögren's Syndrome, but in both cases LF was orally administered.

In this regard, the following information was added to the manuscript:

  • Lines 659-663: “To our knowledge, there are no topical ophthalmic LF products on the market nor clinical studies of topical ophthalmic LF formulations. Only a clinical study in patients with dry eye induced by cataract surgery [107] and a small clinical study conducted on 10 patients with Sjögren's Syndrome [103] but where LF was administered orally has been developed.”
  • Lines 734-741: “Numerous clinical trials are currently ongoing for the use of LF in the fields of cancer, intestinal diseases, and covid-19. Nevertheless, in the ophthalmology field, only one randomized controlled trial about the effect on DED of a combined dietary supplement (for oral administration) containing lactoferrin (among other components such as fish oil, zinc, vitamin C, lutein, vitamin E, γ-aminobutanoic acid and Entero-coccus faecium WB2000) has been reported [149]. Nowadays, there are no clinical trials in progress focused on LF treatment via ophthalmic administration in eyes affected by any of the OSDs mentioned in this review.”

Reviewer 2 Report

This literature review can be described as a comprehensive and descriptive work organised conveniently for the reader. Moreover, the number of sources used to produce this publication is decent. Still, there can be some additions, e.g. it can be mentioned in the section ‘4.1 Role of lactoferrin in the tear fluid’ (rows 203-254) that according to some studies (page 257, Table 2 from Chapter ‘Tear Film’ by J.P. Craig, A. Tomlinson, L. McCann (https://doi.org/10.1016/B978-0-12-374203-2.00229-3) from the ‘Encyclopedia of the Eye’ by Dartt (2010)), an average concentration of lactoferrin in tears is 1.84 mg mL-1.

Also, the following works might be cited in the section ‘5.1 Dry eye and lactoferrin’ (rows 339-377):

1) Selenium-binding lactoferrin is taken into corneal epithelial cells by a receptor and prevents corneal damage in dry eye model animals (2016; https://www.nature.com/articles/srep36903);
2) The relationship between dry eye and lactoferrin levels in tears (2012; Yanwei L, Wei Z, Yu Z. Asian Biomedicine. 2012; https://sciendo.com/it/article/10.5372/1905-7415.0601.130); 

In addition, the following paper on permeation enhancers in ocular drug delivery (Penetration Enhancers in Ocular Drug Delivery, 2019; https://doi.org/10.3390/pharmaceutics11070321) might be cited in rows 477 to 483 of section ‘6. Drug delivery alternatives for the LF topical administration.’

At last, there is an excessive comma after references ‘[56, 57]’ in row 259 on page 7, excessive ellipsis in ‘(LF, cytokines …)’ in row 261 on page 7, and a missing letter ‘l’ in the word ‘Pharmaceutica’ in row 608 on page 15.

In general, this literature review is well-written. It requires minor English language spell checks and is recommended for publication after minor corrections. 

Author Response

Reviewer 2 report:

  1. This literature review can be described as a comprehensive and descriptive work organised conveniently for the reader. Moreover, the number of sources used to produce this publication is decent. Still, there can be some additions, e.g. it can be mentioned in the section‘4.1 Role of lactoferrin in the tear fluid’ (rows 203-254) that according to some studies (page 257, Table 2 from Chapter ‘Tear Film’ by J.P. Craig, A. Tomlinson, L. McCann (https://doi.org/10.1016/B978-0-12-374203-2.00229-3) from the ‘Encyclopedia of the Eye’ by Dartt (2010)), an average concentration of lactoferrin in tears is 1.84 mg mL-1.

We appreciate the reviewer's suggestion. The reference was incorporated in the lines 212.

  1. Also, the following works might be cited in the section ‘5.1 Dry eye and lactoferrin’ (rows 339-377):
    1) Selenium-binding lactoferrin is taken into corneal epithelial cells by a receptor and prevents corneal damage in dry eye model animals (2016;
    https://www.nature.com/articles/srep36903);
    2) The relationship between dry eye and lactoferrin levels in tears (2012; Yanwei L, Wei Z, Yu Z. Asian Biomedicine. 2012;
     https://sciendo.com/it/article/10.5372/1905-7415.0601.130); 

Thanks again for the suggestion. References has been included in line 374-375 and line 386

  1. In addition, the following paper on permeation enhancers in ocular drug delivery (Penetration Enhancers in Ocular Drug Delivery, 2019;https://doi.org/10.3390/pharmaceutics11070321) might be cited in rows 477 to 483 of section ‘6. Drug delivery alternatives for the LF topical administration.’

References has been included in line 515

  1. At last, there is an excessive comma after references ‘[56, 57]’ in row 259 on page 7, excessive ellipsis in ‘(LF, cytokines …)’ in row 261 on page 7, and a missing letter ‘l’ in the word ‘Pharmaceutica’ in row 608 on page 15.

The mistakes and errors have been corrected in the manuscript

  1. In general, this literature review is well-written. It requires minor English language spell checks and is recommended for publication after minor corrections. 

Thanks for the recommendation, we have double-checked the English language

Reviewer 3 Report

The paper can be accepted after major revision ‘Biomedical Applications of Lactoferrin on the Ocular Surface’. The following points can should be discussed

1.       Graphical abstract would be helpful.

2.       What is the uniqueness of this review. If there is already such papers, discuss or cite them.

3.       Cyclodextrin or other nanoparticles are used here for carrying the molecule. Their cavity/porosity will be enough to carry those molecules.

4.       Is there any marketed product? Please cite on clinical trials status.

5.       Would lactoferrin be stable in the outer atmosphere?

6.       How the product based on lactoferrin should be stored?

7.       Discuss on limitations of this therapy.

8.       Why cyclodextrin should be used for carrying molecule. If it is related with solubility, work on some literatures related with solubility imparted by cyclodextrin such as https://doi.org/10.1021/acsomega.0c01228; https://doi.org/10.3390/molecules23051161

Author Response

Reviewer 3 report:

The paper can be accepted after major revision ‘Biomedical Applications of Lactoferrin on the Ocular Surface’. The following points can should be discussed:

  1. Graphical abstract would be helpful.

We appreciate the reviewer's suggestion. The graphical abstract was submitted together with the revised manuscript.

  1. What is the uniqueness of this review. If there is already such papers, discuss or cite them.

In order to clarify the uniqueness of this review, the following information was added to the Concluding Remarks and Future Directions of the revised manuscript:

  • Line 747-752: “More concretely, the uniqueness of this revision in comparison with previous ones includes: the latest update on the topic, the wide scope of LF-related OSDs, the most recent information about the delivery systems for LF ocular topical application, as well as, an easily visualized information through well-crafted illustrations on the LF signaling pathways and the LF beneficial contributions in the mentioned OSDs.”

  1. Cyclodextrin or other nanoparticles are used here for carrying the molecule. Their cavity/ porosity will be enough to carry those molecules.

Reviewer 2 has raised an important point here. The interaction between cyclodextrins and proteins is a topic of ongoing research, as it has potential implications for drug delivery and other biomedical applications. The specific nature of the interaction between cyclodextrins and proteins depends on several factors, including the size and shape of the cyclodextrin, the properties of the protein, and the conditions of the surrounding environment. Some of the possible interactions between cyclodextrins and proteins include: encapsulate proteins within their hydrophobic cavity, protecting them from denaturation and improving their stability and binding to proteins through hydrophobic interactions, hydrogen bonding, and electrostatic interactions.

  1. Is there any marketed product? Please cite on clinical trials status.

The authors sincerely thank the reviewer for their detailed comments. The authors have performed an exhaustive search for information and have not found any details about topical ophthalmic LF products on the market or clinical studies of topical ophthalmic LF formulations. In the reviewed bibliography, the authors have also noticed the existence of a clinical study in patients with dry eye induced by cataract surgery, as well as a small study in Japan conducted in 10 patients with Sjögren's Syndrome, but in both cases LF was orally administered. Numerous clinical trials are currently ongoing for the use of LF in the fields of cancer, intestinal diseases, and covid-19. Nevertheless, in the ophthalmology field, only one randomized controlled trial about the effect on DED of a combined dietary supplement (for oral administration) containing lactoferrin has been reported. In this regard, the following information was added to the manuscript:

  • Lines 659-663: “To our knowledge, there are no topical ophthalmic LF products on the market nor clinical studies of topical ophthalmic LF formulations. Only a clinical study in patients with dry eye induced by cataract surgery [107] and a small clinical study conducted on 10 patients with Sjögren's Syndrome [103] but where LF was administered orally has been developed.”

  • Lines 734-741: “Numerous clinical trials are currently ongoing for the use of LF in the fields of cancer, intestinal diseases, and covid-19. Nevertheless, in the ophthalmology field, only one randomized controlled trial about the effect on DED of a combined dietary supplement (for oral administration) containing lactoferrin (among other components such as fish oil, zinc, vitamin C, lutein, vitamin E, γ-aminobutanoic acid and Entero-coccus faecium WB2000) has been reported [149]. Nowadays, there are no clinical trials in progress focused on LF treatment via ophthalmic administration in eyes affected by any of the OSDs mentioned in this review.”

  1. Would lactoferrin be stable in the outer atmosphere?

An additional challenge in the development of new formulations of LF with the potential to be commercialized is their stability. There are several studies that have assessed the stability of LF in different media and delivery systems. Kim and coworkers studied the stability of LF in solution in the presence of various excipients. They found that arginine and polysorbate 80 can protect the molecule from physical or chemical destabilization. After the addition of arginine, polysorbate 80, or trehalose, it is possible to obtain a solution that remains stable for more than 150 days in the refrigerator. However, stability declines at room temperature or above.

Yao et al. studied the physicochemical stability of liposomes and solid lipid particles loaded with LF and modified by chitosan or pectin. All the LF-loaded liposomes showed a rapid release of the LF, overall, at 40ºC and a complete degradation after a 180-day storage period, whereas almost 30% of LF still remained intact after 180 days into the solid lipid nanoparticles. However, the instability processes in lipid formulation are related to the premature release and expulsion of LF to the environment than to its degradation within the lipid systems.

On the other hand, there is a complete review of the effects of technological treatments used in the treatment and preservation of food on LF. In it can be seen how thermal treatments at high temperatures produce the aggregation and denaturation of LF, as well as some homogenization processes at high pressure when very high-pressure values are used. Other processes, such as spray-drying or freeze-drying, respect the LF structure.

This information has been incorporated in Lines 664-682 of the revised manuscript.

  1. How the product based on lactoferrin should be stored?

The authors thank the reviewer for pointing this out. Based on the author's experience, the best way to preserve and store LF formulations is probably as a re-dispersible freeze-dryer powder.

  1. Discuss on limitations of this therapy.

The main limitation of this therapy is based on the fact that no topical ocular LF products exist currently on the market (Line 659-660). Moreover, it is necessary to take into account the two challenges for designing topical ocular LF formulations: obtaining adequate stability and biocompatibility for the ocular surface administration, which are sometimes hard tasks (this topic is extensively discussed in Lines 664-695 of the manuscript). In addition, as explained in Lines 724 to 727, to understand the effectiveness of the new formulations, it is still necessary to study the in vivo eye pharmacokinetics or the tissue distribution, as well as their response to LF therapy. As it have been added in lines 707-713, LF therapy will constitute a supportive therapy to current therapies, based primarily on reconstituting and reinforcing LF levels to maintain the natural state of the ocular surface.

  1. Why cyclodextrin should be used for carrying molecule. If it is related with solubility, work on some literatures related with solubility imparted by cyclodextrin such as https://doi.org/10.1021/acsomega.0c01228; https://doi.org/10.3390/molecules23051161

The authors really thank the reviewer for such a wise appreciation. Cyclodextrins have another important utility in large peptide or protein molecules, rather than solubilization. It has been shown that cyclodextrins can form complexes with the hydrophobic residues of amino acids, especially aromatic amino acids (phenylalanine, tryptophan, tyrosine, and histidine). Therefore, cyclodextrins can reduce the hydrophobic interactions of proteins by reducing their aggregation and preventing their denaturation. The use of cyclodextrins has been described to promote the unfolding of proteins or to favor the resuspension of its lyophilized products. There are examples of their good behavior with high molecular weight proteins such as some enzymes or monoclonal antibodies.

Reviewer 4 Report

Reviewer Comments: Dear Respected Author, it is a pleasure to accept the task for reviewing your  review article, entitled “Biomedical Applications of Lactoferrin on the Ocular Surface.

The current topic is interesting and I would recommend its publication after a satisfactory round of revision. There are various places where necessary improvements are required. Please consider the following points for improvements.

Please refer to the attached manuscript as well.

1-      Please add more 2 to 3 keywords.

2-      Please add a list of symbols and acronyms before the introduction section to make it more convenient for readers.  

3-      The scope/motivation of the current review is not highlighted/provided. There are already many reviews discussing the same problem.

4-      Please add the actoferrin structure diagram in section 2. It will surely add the value to this section.

5-      In sections 3.3 to 3.5, please add more recent studies discussing the same properties of lactoferrin as mentioned in these sections heading rather than providing/relying on the general information.

6-      Many thanks for preparing Figure 3, it looks good. I tthink the source of information(s) in preparation for Figure 3 should be mentioned/cited in Figure Caption.

7-      Where are the current challenges for the proposed field ?

8-      Just as a bit of advice now trend is shifting towards 3D printing of lactoferrin for biomedical applications. So why not add the para on this topic just before the last heading. It will surely add the value of review. Please consider the following articles on “3D printing”,  “lactoferrin” and “biomedical applications”

i.                    https://doi.org/10.1016/j.msec.2020.111008

ii.                  https://doi.org/10.1016/j.reactfunctpolym.2022.105374

iii.                https://doi.org/10.1016/j.bprint.2022.e00203

iv.                 https://doi.org/10.1016/j.ijbiomac.2022.07.140

v.                 https://doi.org/10.1016/j.ijbiomac.2022.07.140

vi.               https://doi.org/10.3390/pharmaceutics13101698  

9-      Please replace the old references (missing some bibliographies information) in your reference list and replace them with new references. For example, replace old  reference [1], [10], [22], [34] and [71].

Best wishes,

Author Response

Reviewer 4 report:

Reviewer Comments: Dear Respected Author, it is a pleasure to accept the task for reviewing your review article, entitled “Biomedical Applications of Lactoferrin on the Ocular Surface”. The current topic is interesting and I would recommend its publication after a satisfactory round of revision. There are various places where necessary improvements are required. Please consider the following points for improvements. Please refer to the attached manuscript as well.

      1-  Please add more 2 to 3 keywords. 

       As suggested by the reviewer, new keywords have been added in Line 32.

      2-   Please add a list of symbols and acronyms before the introduction section to make it more convenient for readers.  

            The authors thank the reviewer's suggestion, and, following the journal format, a list of acronyms has been added at the end of the reviewed manuscript (before the References section). Lines 770-809.

      3-  The scope/motivation of the current review is not highlighted/provided. There are already many reviews discussing the same problem. 

In order to clarify the scope/motivation of this review, the following information was added to the Concluding Remarks and Future Directions of the revised manuscript:

  • Line 746-752: “In summary, the motivation of this review was to highlight the relevance of LF and its biomedical applications on the ocular surface. More concretely, the uniqueness of this revision in comparison with previous ones includes: the latest update on the topic, the wide scope of LF-related OSDs, the most recent information about the delivery systems for LF ocular topical application, as well as, an easily visualized information through well-crafted illustrations on the LF signaling pathways and the LF beneficial contributions in the mentioned OSDs.”

      4-  Please add the lactoferrin structure diagram in section 2. It will surely add the value to this section. 

           We appreciate the reviewer's suggestion. A representative figure of the LF structure was included in Lines 82-84 of the revised manuscript. [Figure 1]

      5-  In sections 3.3 to 3.5, please add more recent studies discussing the same properties of lactoferrin as mentioned in these sections heading rather than providing/relying on the general information. 

Following the reviewer's recommendations, the authors have performed an updated search for the required information. As a result, the following modifications were carried out:

           In the 3.3 Antibacterial activity section, the following references were added:

  • Line 167: [35] Dierick, M.; Ongena, R.; Vanrompay, D.; Devriendt, B.; Cox, E. Lactoferrin Decreases Enterotoxigenic Escherichia Coli-Induced Fluid Secretion and Bacterial Adhesion in the Porcine Small Intestine. Pharmaceutics 2022, 14. &   [36] Hussan, J.R.; Irwin, S.G.; Mathews, B.; Swift, S.; Williams, D.L.; Cornish, J. Optimal Dose of Lactoferrin Reduces the Resilience of in Vitro Staphylococcus Aureus Colonies. PLoS One 2022, 17.

  • Line 177: [43] Parra-Saavedra, K.J.; Macias-Lamas, A.M.; Silva-Jara, J.M.; Solís-Pacheco, J.R.; Ortiz-Lazareno, P.C.; Aguilar-Uscanga, B.R. Human Lactoferrin from Breast Milk: Characterization by HPLC and Its in Vitro Antibiofilm Performance. Food Sci. Technol. 2022, 59, 4907–4914.   &  [44] Khanum, R.; Chung, P.Y.; Clarke, S.C.; Chin, B.Y. Lactoferrin Modulates the Biofilm Formation and Bap Gene Expression of Methicillin-Resistant Staphylococcus Epidermidis. Can. J. Microbiol. 2023, 69.

        In the 3.4 Antiviral and antifungal activity section, the following references were included:

  • Line 186: [48] Mancinelli, R.; Rosa, L.; Cutone, A.; Lepanto, M.S.; Franchitto, A.; Onori, P.; Gaudio, E.; Valenti, P. Viral Hepatitis and Iron Dysregulation: Molecular Pathways and the Role of Lactoferrin. Molecules 2020, 25. [49] Picard-Jean, F.; Bouchard, S.; Larivée, G.; Bisaillon, M. The Intracellular Inhibition of HCV Replication Represents a Novel Mechanism of Action by the Innate Immune Lactoferrin Protein. Antiviral Res. 2014, 111, 13–22.   &    [50] Wakabayashi, H.; Oda, H.; Yamauchi, K.; Abe, F. Lactoferrin for Prevention of Common Viral Infections. Infect. Chemother. 2014, 20, 666–671.

  • Line 188: [52] Machado, R.; da Costa, A.; Silva, D.M.; Gomes, A.C.; Casal, M.; Sencadas, V. Antibacterial and Antifungal Activity of Poly(Lactic Acid)-Bovine Lactoferrin Nanofiber Membranes. Biosci. 2018, 18. &  [53] Pawar, S.; Markowitz, K.; Velliyagounder, K. Effect of Human Lactoferrin on Candida Albicans Infection and Host Response Interactions in Experimental Oral Candidiasis in Mice. Arch. Oral Biol. 2022, 137.

              In the 3.5 Anticarcinogenic properties section, the following references were added:

  • Line 195: [54] Rocha, V.P.; Campos, S.P.C.; Barros, C.A.; Trindade, P.; Souza, L.R.Q.; Silva, T.G.; Gimba, E.R.P.; Teodoro, A.J.; Gonçalves, R.B. Bovine Lactoferrin Induces Cell Death in Human Prostate Cancer Cells. Med. Cell. Longev. 2022, 2022.  [55] Arcella, A.; Oliva, M.A.; Staffieri, S.; Alberti, S.; Grillea, G.; Madonna, M.; Bartolo, M.; Pavone, L.; Giangaspero, F.; Cantore, G.; et al. In Vitro and in Vivo Effect of Human Lactoferrin on Glioblastoma Growth. J. Neurosurg. 2015, 123, 1026–1035.   [56] Ramírez-Sánchez, D.A.; Arredondo-Beltrán, I.G.; Canizalez-Roman, A.; Flores-Villaseñor, H.; Nazmi, K.; Bolscher, J.G.M.; León-Sicairos, N. Bovine Lactoferrin and Lactoferrin Peptides Affect Endometrial and Cervical Cancer Cell Lines. Biochem. Cell Biol. 2021, 99, 149–158.  &   [57] Nakamura-Bencomo, S.; Gutierrez, D.A.; Robles-Escajeda, E.; Iglesias-Figueroa, B.; Siqueiros-Cendón, T.S.; Espinoza-Sánchez, E.A.; Arévalo-Gallegos, S.; Aguilera, R.J.; Rascón-Cruz, Q.; Varela-Ramirez, A. Recombinant Human Lactoferrin Carrying Humanized Glycosylation Exhibits Antileukemia Selective Cytotoxicity, Microfilament Disruption, Cell Cycle Arrest, and Apoptosis Activities. Invest. New Drugs2021, 39, 400–415.

      6-  Many thanks for preparing Figure 3, it looks good. I think the source of information(s) in preparation for Figure 3 should be mentioned/cited in Figure Caption. 

We thank the reviewer for his/her kind comments. In line with this suggestion, the sources of information have been included in Figure 3. [currently Figure 5 in the revised manuscript, Line 494].

      7-  Where are the current challenges for the proposed field?

The authors thank the reviewer for pointing this out. Based on the author's experience and the extended review of the bibliography, the current challenges for the proposed field include the creation of a biocompatible topical ocular formulation of LF that allows adequate permanence on the ocular surface and that can even be compatible with existing treatments. This information has been incorporated in Lines 742-744 of the revised manuscript.

      8-  Just as a bit of advice now trend is shifting towards 3D printing of lactoferrin for biomedical applications. So why not add the para on this topic just before the last heading. It will surely add the value of review. Please consider the following articles on “3D printing”,  “lactoferrin” and “biomedical applications”

  1. https://doi.org/10.1016/j.msec.2020.111008
  2. https://doi.org/10.1016/j.reactfunctpolym.2022.105374

            iii.     https://doi.org/10.1016/j.bprint.2022.e00203 

  1. https://doi.org/10.1016/j.ijbiomac.2022.07.140
  2. https://doi.org/10.1016/j.ijbiomac.2022.07.140
  3. https://doi.org/10.3390/pharmaceutics13101698  

The authors thank the reviewer for the excellent suggestion. The authors have made a review of the available information and have added information on the suggestion given by the reviewer in the new version of the manuscript:

  • Lines 683-695: “In recent years, 3D printing has emerged as a promising technology for creating complex structures and materials with precise control over their physical properties [146]. There has been some research into 3D printing lactoferrin, particularly for its potential use in biomedical applications but not specifically for ophthalmic applications. One example is the 3D printing of lactoferrin-loaded alginate hydrogel scaffolds using a bioprinter [147]. The researchers found that the printed scaffolds had good biocompatibility and could support the growth of human mesenchymal stem cells. Another example is the development of a 3D-printed lactoferrin-based hydrogel that could be used as a wound dressing [148]. The researchers found that the hydrogel had good mechanical properties and could release lactoferrin in a controlled manner. Overall, 3D printing lactoferrin holds promise for the development of new therapeutic and biomedical applications. However, further research is needed to fully understand the potential of this technology and to optimize the printing process of lactoferrin-based hydrogels for ophthalmic applications.”

      9-  Please replace the old references (missing some bibliographies information) in your reference list and replace them with new references. For example, replace old reference [1], [10], [22], [34] and [71].

In line with the reviewer's suggestion, the proposed references were updated.

  • Reference [1] (Sorensen, M.; Sorensen, S.P.L. The proteins in whey. Compte Rendu Des. Trav. Lab. Carlsberg Ser. Chim. 1940, 23, 55–99), was replaced with [1] (Groves, M.L. The Isolation of a Red Protein from Milk. Am. Chem. Soc. 1960, 82, 3345–3350, doi:10.1021/JA01498A029).
  • Reference [8] (Anderson, B.F.; Baker, H.M.; Dodson, E.J.; Norris, G.E.; Rumball, S.V.; Waters, J.M.; Baker, E.N. Structure of human lactoferrin at 3.2-A resolution. Natl. Acad. Sci. U.S.A. 1987, 84, 1769-73; DOI: 10.1073/pnas.84.7.1769), was replaced with the new reference [8] (Wally, J.; Buchanan, S.K. A Structural Comparison of Human Serum Transferrin and Human Lactoferrin. Biometals 2007, 20, 249–262, doi:10.1007/S10534-006-9062-7).
  • Reference [10] has been removed.
  • Reference [22] (Shau, H.; Kim, A.; Golub, S.H. Modulation of natural killer and lymphokine-activated killer cell cytotoxicity by lactoferrin. J. Leukoc. Biol. 1992, 51, 343-9), was replaced with the new reference [21] (Shi, H.; Li, W. Inhibitory Effects of Human Lactoferrin on U14 Cervical Carcinoma through Upregulation of the Immune Response. Lett. 2014, 7, 820–826; DOI:10.3892/OL.2013.1776).
  • Reference [34] (Crouch, S.P.; Slater, K.J.; Fletcher, J. Regulation of cytokine release from mononuclear cells by the iron-binding protein lactoferrin. 1992, 80, 235-40), was replaced with the new reference [33] (Li, H.Y.; Yang, H.G.; Wu, H.M.; Yao, Q.Q.; Zhang, Z.Y.; Meng, Q.S.; Fan, L.L.; Wang, J.Q.; Zheng, N. Inhibitory Effects of Lactoferrin on Pulmonary Inflammatory Processes Induced by Lipopolysaccharide by Modulating the TLR4-Related Pathway. J. Dairy Sci. 2021, 104, 7383–7392, doi:10.3168/JDS.2020-19232).
  • Reference [71] (Franklin, R.M.; Kenyon, K.R.; Tomasi, T.B. Immunohistologic Studies of Human Lacrimal Gland: Localization of Immunoglobulins, Secretory Component and Lactoferrin. Immunol. 1973, 110.), now numbered [78] due to bibliography additions, has been maintained as the authors believe that it reveals valuable information on the secretion of lactoferrin by the lacrimal gland. However, the reference number [79] (Singh, S.; Ali, M.J.; Mittal, V.; Brabletz, S.; Paulsen, F. Immunohistological Study of Palpebral Lobe of the Lacrimal Gland in Severe Dry Eyes Secondary to Stevens-Johnson Syndrome. Curr. Eye Res. 2021, 46, 789–795) has been added because it provides updated value on this topic.

Round 2

Reviewer 1 Report

The authors have improved the manuscript and solved my previous concerns. In this case, I would like to recommend the acceptance of this revised version.

Author Response

Many thanks to the reviewer for his work that has improved our manuscript.

Reviewer 3 Report

Hydrophobic interaction of drug/molecule with cyclodextrin is well mentioned on these artciles; update your manuscript. write some discussions on it. 

https://doi.org/10.1021/acsomega.0c01228; https://doi.org/10.3390/molecules23051161

Author Response

Thank you very much for the interesting suggestion of the reviewers. We have included the following text in the manuscript (from line 684) and the following references (in green in the text):

CDs can be a candidate to contribute to the stabilization of the LF in the nanomedicine and hydrogels. CDs are cyclic oligosaccharides that show a good ability to form complexes with drug molecules and to improve their physicochemical properties without molecular modifications, via drug/host interaction [149, 150]. The capacity of CDs to interact with proteins is well known [151-155]. Different mechanisms have been described by which cyclodextrins interact with proteins improving physical and chemical stability [154]. CDs can form inclusion complexes with amino acids, mainly βCD derivatives and hydrophobic and aromatic residues of Phe, Tyr, His, and Trp, modulating the solvent exposure to hydrophobic amino-acidic residues [151-154]. Additionally, the surface activity of some CD-derivative can contribute to protein stabilization by reducing the protein surface adsorption [154]. DCs can prevent protein aggregation and adsorption through these mechanisms and improve stability against proteases. Consequently, CDs derivatives are excellent candidates for improving the chemical and physicochemical stability of proteins in the solid and liquid states [155]. 

  1. Saokham, P.; Muankaew, C.; Jansook, P.; Loftsson, T. Solubility of Cyclodextrins and Drug/Cyclodextrin Complexes. Molecules 2018, 23, 1161. https://doi.org/10.3390/molecules23051161
  2. Kovacs, T.; Nagy, P.; Panyi, G.; Szente, L.; Varga, Z.; Zakany, F. Cyclodextrins: Only Pharmaceutical Excipients or Full-Fledged Drug Candidates? Pharmaceutics 2022, 14, 2559. https://doi.org/10.3390/pharmaceutics14122559
  3. Irie, T., Uekama, K., Cyclodextrins in peptide and protein delivery. Adv Drug Deliv Rev. 1999 1;36(1):101. doi: 10.1016/s0169-409x(98)00057-x.
  4. Härtl E, Winter G, Besheer A. Influence of hydroxypropyl-Beta-cyclodextrin on the stability of dilute and highly concentrated immunoglobulin g formulations. J Pharm Sci. 2013 Nov;102(11):4121-31. doi: 10.1002/jps.23729.
  5. Castañeda Ruiz, A.J.; Shetab Boushehri, M.A.; Phan, T.; Carle, S.; Garidel, P.; Buske, J.; Lamprecht, A. Alternative Excipients for Protein Stabilization in Protein Therapeutics: Overcoming the Limitations of Polysorbates. Pharmaceutics 2022, 14, 2575. https://doi.org/10.3390/pharmaceutics14122575
  6. Stolzke, T.; Krieg, F.; Peng, T.; Zhang, H.; Häusler, O.; Brandenbusch, C. Hydroxylpropyl- -cyclodextrin as Potential Excipient to Prevent Stress-Induced Aggregation in Liquid Protein Formulations. Molecules 2022, 27, 5094. https://doi.org/10.3390/molecules27165094155.

155         Serno, T.; Geidobler, R.; Winter, G.; Protein stabilization by cyclodextrins in the liquid and dried state. Adv Drug Deliv Rev. 2011 63(13):1086. doi: 10.1016/j.addr.2011.08.003.

Reviewer 4 Report

Authors have revised their article and incorporated all of the reviewers suggestions.  Hence I would recommend its publication in the current form.
Best wishes,

Author Response

Many thanks to the reviewer for his work that has contributed to improving our manuscript.